# Empowering Non-Academic Staff for the Implementation of Sustainability in Higher Education Institutions

Paula Bacelar-Nicolau [1,2,]*, Mahsa Mapar [1,3], Sandra Caeiro [1,3], Sara Moreno Pires [4], Mariana Nicolau [4], Catarina Madeira [4], Marta Ferreira Dias [5], Ana Paula Gomes [6], Myriam Lopes [6], Helena Nadais [6] and Georgios Malandrakis [7]

1 Centro de Estudos Globais, Departamento de Ciência e Tecnologia, Universidade Aberta, 1269-001 Lisboa, Portugal; m.mapar@fct.unl.pt (M.M.); scaeiro@uab.pt (S.C.)
2 CFE-Centre for Functional Ecology, Universidade de Coimbra, 3040-256 Coimbra, Portugal
3 CENSE-Center for Environmental and Sustainability Research & CHANGE—Global Change and Sustainability Institute, NOVA School of Science and Technology, NOVA University Lisbon, Caparica, 1099-085 Lisboa, Portugal
4 GOVCOPP-Research Unit on Governance, Competitiveness and Public Policies, Department of Social, Political and Territorial Sciences, University of Aveiro, 3810-193 Aveiro, Portugal; sarapires@ua.pt (S.M.P.); mariananicolau@ua.pt (M.N.); catarinabmadeira@ua.pt (C.M.)
5 GOVCOPP-Research Unit on Governance, Competitiveness and Public Policies, Department of Economics, Management and Industrial Engineering and Tourism, University of Aveiro, 3810-193 Aveiro, Portugal; mfdias@ua.pt
6 CESAM-Centre for Environmental and Marine Studies, Department of Environment and Planning, University of Aveiro, 3810-193 Aveiro, Portugal; pgomes@ua.pt (A.P.G.); myr@ua.pt (M.L.); nadais@ua.pt (H.N.)
7 Department of Education, Aristotle University of Thessaloniki, 541 24 Thessaloniki, Greece; gmalandrakis@eled.auth.gr
* Correspondence: pnicolau@uab.pt

**Abstract:** Sustainability within higher education institutions (HEIs) is a well-established topic in the literature. Many fields of education for sustainable development have been explored, mainly focused on HEI students, as well as on academic staff. The technical, administrative, and management staff, referred to as non-academic staff has not received as much attention as the remaining HEI community, which leaves a gap in the successful implementation of sustainability practices and policies, as they play a vital and central role in the HEIs' everyday functioning. Hence, the EUSTEPs project launched two sustainability training courses dedicated exclusively to this segment of the university community, aiming to increase their knowledge on facts and tools for the best sustainability transition. The first short-term online training, organized by the University of Aveiro and Universidade Aberta, Portugal, was run in May 2021. The training targeted 27 non-academic staff from different sectors. The second online training course was implemented one year later and involved 17 elements from the previous training. The results showed very high levels of overall satisfaction and full achievement of the participants' expectations in sustainability issues. The non-academic staff learned and discussed the human–environment relationship, tracked and discussed their personal ecological footprint in the workspace, actively participated on how to run the university ecological footprint calculator, developed within the EUSTEPs project, and felt mobilized to implement actions to reduce their university's environmental impacts (as well as in their general daily activities). Similar training programs can be used to empower non-academic staff for the implementation of sustainability in other higher education institutions, hence contributing to a successful integrated sustainability approach for the whole school.

**Keywords:** non-academic staff; environmental sustainability; ecological footprint; Portugal

## 1. Introduction

Higher education institutions (HEIs) are a labor-intensive sector in which a wide range of academic and non-academic staff and students engage in various institutional activities within the wider world. While academic staff primarily carry out teaching, research, and outreach activities, HEIs also rely on the support of non-academic staff to ensure the strategic, technical, administrative, financial, and operational aspects of all these activities [1]. On the other hand, HEIs are generally considered significant contributors to the promotion of sustainability, as they are catalysts for the development of knowledge towards sustainable development for present and future citizens, both in their work and personal environments [1]. HEIs congregate a great number of individuals, from the student population to the teachers and administrators, and hence become the main stage for the dissemination and implementation of sustainability concepts and behaviours s. Designing a sustainable university requires significant involvement of the wider academic community, including technical, administrative and management staff, who have the authority and power to make decisions for and against sustainability initiatives and practices in the short, medium, and long term. Also, their empowerment motivates them to become sustainability proponents and leaders [2]. However, there is still little evidence regarding specific training on sustainability across the technical, administrative, and management staff, and even less regarding their impact on promoting sustainability at HEIs. Hence, it becomes crucial to involve them in the perspective of their HEI's transition towards sustainability.

Designed to reach the entire community at an HEI, the ERASMUS+ EUSTEPs project—Enhancing Universities' Sustainability Teaching and Practices through Ecological Footprint—has been set up through a strategic partnership among four European universities and an international non-governmental organization to engage the whole university community (students, academic, and non-academic staff) in sustainability education and practice. Through this wide-range scope of action, the project aimed to refresh and innovate sustainability teaching and practices in HEIs using the ecological footprint methodology. This project focused on one target group at a time—it started with a module dedicated exclusively to students, followed by training devoted to university educators and PhD students. Whilst these groups' training modules yielded encouraging results, it was time to bring the project's pedagogies to the non-academic staff members of the HEI. The EUSTEPs project also developed a massive online open course (MOOC), available to anyone in the general public that has interest in learning further about the complexity of sustainability. Alongside the pedagogical materials, the project developed and presented the university footprint calculator—a free tool available online to all HEIs that wish to measure and monitor their ecological footprint [3]. All the information about the project is available from the website: http://www.eusteps.eu (accessed on 3 October 2023).

The main aim of this paper is to assess the implementation, learning outcomes, and main results of the training courses targeting the technical, administrative, and management staff, from now on referred to as "non-academic staff", of two Portuguese universities involved in the partnership. Specifically, it aims to understand the impact of these training courses on non-academic staff, with a focus on: (i) making them more aware of sustainability and ecological footprint (EF) concepts, and (ii) empowering them to transform their workspace and their university community towards becoming more sustainable.

This paper is structured as follows: the first section discusses the critical role of non-academic staff in the overall running and performance of HEIs and the success of sustainability changes within these institutions, despite being mostly forgotten by the literature on education for sustainable development. The subsequent section presents the methodology of the study, addressing the proposed goals and describing the process of structuring the sustainability and ecological footprint training courses conducted for these staff within two Portuguese Universities, the training itself, and the training assessment tools. The results and discussion sections present the sample and analyze the main outcomes. The last section outlines the core conclusions, focusing on the goals of the training courses as well as providing recommendations and suggestions for further work.

## 2. Review of Sustainability Implementation with Non-Academic Staff

Sustainable development principles involve inducing changes and transitions towards a society that has balanced the environmental, social, economic, and institutional dimensions of its daily activities. HEIs have an increasing responsibility as agents in promoting sustainable development principles by creating knowledge, transferring this knowledge to society, and preparing students for their future role in society [4]. Given their wide scope of action and the broad spectrum of actors involved, HEIs hold a strategic and prominent position, specifically when they are public institutions, to approach and address sustainability [5]. Throughout the last decades, several commitments and initiatives have been promoted and signed worldwide, with the main aim to ensure that sustainable development principles are integrated into the main dimensions and fields of action of HEIs, namely education, research, campus operations, outreach, assessment, and reporting [6,7]. The success of a sustainability plan, strategy, or policy in each HEI relies on the commitment of the entire community—not only those in charge of the institution, but also students, academic, and non-academic staff—which outlines the importance of the contribution and involvement of every individual inside campuses [8]. Saito et al. [9] underlined the importance of "*a more collaborative governance approach within higher education for Higher Education for Sustainable Development*" [9] (pp. 1640). Also, Leal Filho et al. [10] argued that HEIs that adopt sustainability policies as instruments of governance will most likely witness a greater engagement of their staff. In Portugal, there is no formal law or regulation at the governmental level advocating for change towards sustainability in HEIs [11]. To overcome this issue, the sustainable campus network *Rede Campus Sustentável* (RCS) was recently established, aiming to increase action and collaboration between Portuguese HEIs to allow for advanced sustainability implementation within Portuguese HEIs [12].

Overall, non-academic staff of HEIs are becoming widely recognized as critical elements for the success of sustainability changes in HEIs, not only because of their support on administrative, management, and technical issues for the whole university community, but also due to their "technical expertise" that serve as catalysts for sustainable revolution at an institution [13–15].

Non-academic staff have become more qualified in recent decades [13]. A trained and skilled staff plays an important role in the overall running of an HEI, safeguarding efficiency and success in the support of all the activities that occur in these institutions [16]. Therefore, it becomes vital to ensure proper training and development opportunities for this staff in order to assure they are equipped with tools that allow them to carry out the tasks and challenges proposed by the institutional environment. The process of lifelong learning is crucial to every individual in their workplace [17,18]. Nevertheless, the literature has been mainly focused on the role of academic staff for sustainability in HEIs, leaving a gap in the importance that the non-academic body has on the overall performance of these institutions [19].

To empower and inspire non-academic staff to participate and engage in sustainability change and initiatives in their institutions, their knowledge and skills in sustainable development need strengthening. Non-academic staff should be involved as part of a holistic approach to education for sustainable development (EDS) within HEIs [20,21]. Acquiring knowledge in the sustainable development sphere is a central piece in the implementation and dissemination of sustainable practices [22]. A study at the University of Aegean, Greece, showed that non-academic staff, when not properly informed and trained on sustainability issues, may not be as involved in sustainability initiatives promoted by their university [8]. Also, the same participants indicated their willingness to engage in and adopt more sustainable practices, both as individuals and at the institutional level, but felt unconfident in their level of knowledge regarding this issue [8]. Dabija et al. [23] also highlighted the prominence of non-academic staff training as a key part of a national strategy in Romania for HEIs to pursue sustainable paths. Training on sustainable development needs to be focused not only on the basic concepts of the issue but should also be in line with the institutional context of HEI and be endorsed by an institutional strategy

that favors sustainability [24]. Hence, there is a need for research on strategies to approach these communities and evaluate their effectiveness to promote sustainability.

### 3. The EUSTEPs Project and the Universities Involved

The EUSTEPs project—Enhancing Universities' Sustainability Teaching and Practices—was a European project (2019–2022) funded by the ERASMUS+ program that arose from a strategic partnership between the Aristotle University of Thessaloniki (AUTH), the coordinating institution located in Greece; the University of Aveiro (UAV) and Universidade Aberta (UAb), both situated in Portugal; the University of Siena (UNISI) in Italy; and the international non-governmental organization Global Footprint Network (GFN) based in California (USA).

EUSTEPs aimed to introduce and educate the European academic community on sustainability, employing a broader and holistic approach to tackle the most urgent issues and topics that currently impair sustainable development. The project proposed a "learning by doing" approach to raise awareness of sustainability system thinking and educate the wider academic and non-academic community on the basics of the concept of sustainability, ecological overshoot, and the Sustainable Development Goals (SDGs) of the United Nations (UN) Agenda 2030, through the lenses of the ecological footprint (EF). The project targeted the whole HEI community, including (i) students, (ii) academic staff (e.g., professors, educators, researchers), and (iii) non-academic staff (e.g., technical, administrative, and management staff), by adopting an experimental and practical approach to sustainability teaching, and learning, with the final goal of contributing to a new generation of citizens and professionals who are literate on sustainability issues.

The EUSTEPs project developed the university footprint calculator, aiming to provide a tailored online tool for assessment, monitoring, and management of higher education institutions' consumption of natural resources and ecosystem services based on a standardized ecological footprint methodological approach. After several months of fine-tuning and testing, the Calculator was made available on the project's official website, online and free of charge, to any HEI interested in understanding its impact on the environment (Galli et al., unpublished work) [25].

In the first step of the project, the students' module was developed and launched in the spring of 2020 as a pilot in the four Universities involved in the project. In the second step, based on the outcomes of the students' pilot teaching, and after refining the module, short-term online training, "EUSTEPs Module: Educators' and Ph.D's Online Training", was carried out in September 2020, targeting both academic educators and Ph.D. students, who were trained on how to teach the EUSTEPs module in a cross-cutting and interactive way and how to build an Ecological Footprint calculator for their HEIs, respectively. Details of the student and educators' teaching modules were published in Moreno Pires et al. [3] and Malandrakis et al. [26], respectively.

As a third step, the project intended to develop sustainability and ecological footprint training specifically targeting non-academic staff of the four universities. For language reasons, both Portuguese universities in the project, UAb and UAV, opted for developing the training courses in Portuguese, facilitating communication and interaction between participants in the same institution, as well as giving the opportunity to interact and discuss among members of the two institutions.

Both universities are relatively young in Portugal and have recently been highly engaged in implementing sustainability practices on their campus. While the University of Aveiro was founded in 1973, the Universidade Aberta was founded in the following decade, in 1988. UAb is also the only Portuguese public distance education university, which includes both undergraduate and graduate higher education courses and Lifelong Learning, that is dedicated to students from all Portuguese-speaking countries. Table 1 summarizes some information about UAV and UAb and their relevant sustainability initiatives.

**Table 1.** UAb and UAV information.

| | UAb | UAV |
|---|---|---|
| Is the HEI a single or multi-campus university? | Single | Multi |
| If it is a multi-campus university, are all the campuses located in the same city or in different cities? | Dellegations in different cities | Different cities |
| Is the HEI distance learning or face- to-face? | Online | Face to face |
| Area ($m^2$) | 12,948 | 1,500,000 |
| Educational offer (2021) | 11 undergraduate programs, 22 master programs, 9 post-graduate programs, and 10 doctoral programs | 45 undergraduate courses, 77 graduate programs, and 51 doctoral programs |
| Total number of students enrolled (2021) | 7000 | 15,000 |
| Number of educators | 150 | 1000 |
| Number of non-academic staff (2021) | 180 | Approximately 700 |
| Number of annual graduations (2019) | 434 | 4089 |
| Does a sustainability office exist? | No | No |
| If no, does a best practice group (or other) exist? (2022) | Yes, a working group (WG) called 'Sustainable Campus', Universidade Aberta, was established in 2021. | Yes, UAV had a mission group for sustainable development, but it is setting up a new "Sustainability Forum" for 2023. |
| Office (or group) staffing and membership (2022) | The WG functions under the direct dependency of the rector and consists of 11 main members, including representatives of the rectoral team, teaching departments and research units, administrative staff, and former students. | The "Sustainability Forum" will have one representative of the Rectoral Team, Administration, Social Action Services, Students, Pedagogical Council, Scientific Council, Organic Units, and Research Units. |
| Main functions of the Sustainability Office (or best practice group) (2022) | The main functions of the WG are to promote sustainability issues within UAb, contribute to a more sustainable society through the identification of measures and good practices for sustainability at UAb, and propose strategies and measures to achieve the Sustainable Development Goals. | It is a consultative forum. Functions and competences are to be regulated after February 2023. |
| Production of a sustainability report (2022) | No formal sustainability report produced, but published the sustainability tracking, assessment, and rating system report in 2019. | First sustainability report published in 2022. |
| Presence of an Ecological Footprint Assessment | Within the EUSTEPs project, a footprint assessment was conducted for 2019–2020 for consumption activities under the direct responsibility of the university administration. Areas of indirect responsibility were not tracked. | Within the EUSTEPs project, a footprint assessment was conducted for 2018–2019 for consumption activities under the direct responsibility of the university administration. Areas of indirect responsibility were not tracked. |

## 4. Methods

### 4.1. Short-Term Sustainability Training Courses for Non-Academic Staff

Both universities worked jointly to design two short-term sustainability and ecological footprint training courses for non-academic staff in the Portuguese language based on the materials and methods prepared by all the partners within the EUSTEPs project. Table 2 summarizes the main information about these courses, and the next sections will detail their goals and structure.

**Table 2.** Information about 1st and 2nd training sessions.

| | **First Short-Term Training** | **Second Short-Term Training** |
|---|---|---|
| Training name | "Sustainability Training for University non-academic staff" | "Sustainability Training for University non-academic staff: the University Footprint Calculator" |
| Main goal of the training | The concepts of Sustainability and the Ecological Footprint within HEIs. | Presentation of the final product of EUSTEPs, including EUSTEPs' university footprint calculator. |
| Date of training | May 2021 | May 2022 |
| Duration | 10.5 h | 3.5 h |
| Target audience | Non-academic staff | Non-academic staff |
| Number of participants | UAb-13 UAV-14 | UAb-7 UAV-10 |
| Questionnaires applied to the participants | Pre-questionnaire to evaluate the participants' pre-knowledge and post-questionnaire to evaluate their perceptions of the training and their overall feedback and satisfaction after the course. Post-questionnaire to evaluate the participants' perceptions of the training and their overall feedback and satisfaction after the course. | Post-questionnaire to evaluate the participants' perceptions of the training and their overall feedback and satisfaction after the course. |

The first training course, with the participation of 27 non-academic staff, mainly served as a general course to familiarize the staff with the concepts of sustainability and ecological footprints, as well as how to reduce their personal ecological footprint in their workplace and activities. However, the main goal of the second training course was to expand the details of the university footprint calculator and the data gathering phase. Therefore, only the administrative staff who were later involved in data gathering for the calculation of the university ecological footprint were enrolled, and the second training course covered 17 administrative staff from both universities. The methodological approach in both training courses was designed with careful consideration of the requirements and objectives of the EUSTEPs project to enable the non-academic staff to directly work with the university footprint calculator while also imparting sustainability through the lens of ecological footprints.

4.1.1. First Training Course

Since UAb is a distance learning University and it has its own e-learning platform, and given the context of the COVID-19 pandemic in 2020, the teams worked together to design an online training course. The short-term online joint training course was entitled "Sustainability Training for University non-academic staff" and was designed for the non-academic staff of both Portuguese universities.

The course was designed to tackle the broad spectrum of sustainability—including concepts of sustainability and the ecological footprint within HEIs—grouping activities and moments of individual learning. The structure of the training course is shown in Table 3. The content of the training course was divided into four main topics: (i) sustainability, ecological goals, overshoot, and Sustainable Development Goals; (ii) ecological footprint and sustainability within everyday life; (iii) personal ecological footprint, and (iv) higher education institutions and sustainability.

**Table 3.** Training course structure with the course objectives, content, and the applied educational materials and activities.

| Topic | Learning Path | Length (hours) | Learning Objectives | Educational Materials | Activities |
|---|---|---|---|---|---|
| T1. Sustainability, ecological overshoot, and SDGs | Synchronous and asynchronous | 2.5 h | • Sustainability concept and main aspects within the institutional context<br>• Ecological overshoot concept, ecosystem limits<br>• Importance of knowledge and cooperation for sustainability<br>• SDGs | • Videos<br>• Slides<br>• Quiz<br>• Further readings | • Pre-questionnaire<br>• Group activity (game): Fisherman for one hour (synchronous) |
| T2. EF and sustainability within everyday life | Asynchronous | 3.5 h | • EF: concept, measurement units, factors, and utility<br>• EF usefulness as an SD indicator<br>• Relationship between SDGs and EF | • Videos<br>• Slides<br>• Forum<br>• Further readings | • Group activity: daily activities at work and the impact of EF on campus<br>• Forum discussion about their institutions |
| T3. Personal EF | Asynchronous | 1 h | • Accounting personal EF<br>• Gaps between EF and natural resources availability<br>• Possible solutions to reduce personal EF<br>• Assess personal impact on the planet | • Slides<br>• Forum<br>• Further readings | • Individual activity: calculating personal EF<br>• Forum discussion about personal EF |
| T4. HEIs and Sustainability | Synchronous and asynchronous | 3.5 h | • Sustainability assessment tools for HEIs<br>• Different aspects of the HEI's sustainability<br>• Assess the dimensions of sustainability within the institution<br>• Main parameters in the University EF Calculator | • Videos<br>• Slides<br>• Forum | • Post-questionnaire<br>• Group activity: how to improve sustainability at my university?<br>• Group debate: main parameters to address in the calculator (forum and synchronous session) |

Different educational materials, such as learning guidelines, videos, slides, quizzes, group e-activities, and guided debate forums were used for each topic. The course developed on the e-learning platform of Universidade Aberta mainly involved an asynchronous learning path including two synchronous moments. In total, the participants were expected to be involved for 10.5 h (see Table 2). The course was active for 11 working days (10.5 h), from 4–17 May 2021.

The approach in the first round of training deliberated both synchronous and asynchronous learning paths, since knowledge about theoretical approaches to sustainability and ecological footprints could be improved to a greater extent through the synchronous method, whereas the asynchronous method can result in greater improvements in skills related to communication and e-learning [27].

The course participants were invited to enroll in asynchronous learning activities, studying the provided lessons (which included learning texts, videos, and slide shows, with optional reading materials for further learning), completing the proposed activities (individual quizzes and guided group activities), and engaging in the guided debate forums (see Table 3). The two synchronous sessions included: (i) a 1 h session at the beginning of the training as an ice breaker for the participants of both universities and the trainers, and (ii) a 1 h session at the end of the training to present and discuss the results of the group activities as well as the next steps of the project.

The first topic of the training, *Sustainability, EF, and SDGs* was introduced in the first synchronous session using the *Fish Game*, a group activity designed to introduce one of the most basic concepts of sustainability: ecosystems limits. After the synchronous session, the participants were guided to follow a lesson on the e-learning platform, which approached the concepts of sustainability and ecological overshoot, as well as the understanding of the importance of knowledge and cooperation towards sustainability and the 2030 Agenda. The Moodle e-learning lesson included a study guide with learning objectives, slides, and

educational videos intertwined with short quizzes on the fundamental concepts and optional further readings for those who wished to deepen their knowledge on the topic. Short breaks were also included in the Moodle e-learning lesson (i.e., short physical activities) after a set of slides/videos or e-activities. Hence, the Moodle lesson was structured using gamification elements [28].

The second topic, *EF and Sustainability within everyday life* presented the concept of the EF as well as its measurement unit, factors, and utility as a tool for environmental accountability and sustainability indicators. In addition to a second e-learning lesson (including learning objectives, a slide show, videos, quizzes, stretching pauses, and further readings), a group e-activity was carried out whereby each group identified activities/behaviours at their own university with a high impact on the institution's EF. Additionally, proposals for improvement were presented in a debate forum.

The third topic, *Personal EF* included an individual activity in which each participant calculated his/her personal EF using the EF calculator available on the Global Footprint Network website (Global Footprint Network. Ecological Footprint Calculator. Available at: https://www.footprintcalculator.org/home/en). The participants then shared their results on a communal Google Form. After the calculation, the participants were asked to reflect on and discuss the results in a debate forum moderated by the trainers.

The last topic, *HEI and Sustainability* addressed the sustainability assessment tools in HEIs and the core elements of sustainability at HEIs, as well as presented some initiatives towards sustainability taking place in other institutions. This was also presented as an e-learning lesson (including learning objectives, slides, videos, quizzes, and a stretching pause). The course participants were also introduced to the first version of the EF calculator for universities developed by the EUSTEPs project, and they analyzed the categories used in the EF calculator. In the group e-activity associated with this topic, each group of participants chose one of the main categories analyzed using the university EF calculator (i.e., energy, mobility, water and wastewater, buildings, and electronics and equipment) and then identified the relevant SDGs for their selected category and proposed sustainable solutions to be implemented within the campus for that category. The results of the group e-activity were presented and discussed in the second synchronous session, which took place on the last day of the training course. Also, the individual results of the EF calculator (obtained by the participants in the third topic) were presented and discussed during this synchronous session. This last synchronous session served both to outline the main outcomes of the training on sustainability at universities (for all parts involved) and as a reflective moment for the participants to share their insights as well as perceived strengths and opportunities for the development of the course.

4.1.2. Second Training Course

The second short online training course for the non-academic staff took place on 3 May 2022, as a follow up to the earlier training, with the main objective of presenting and discussing the EUSTEPs university footprint calculator. It was crafted to be a 3 h and 30 min online workshop. This was done purposefully, as the synchronous nature of the workshop, applied during the second round, allowed for a direct exchange of ideas and fostered collaborative work skills [29]. This is particularly relevant when applying the university footprint calculator, as it requires the co-participation of several staff members and departments. Therefore, the collaborative aspect of the workshop was highly important for the successful application of the calculator at the university.

The workshop was structured as follows: (i) presentation of the EUSTEPs main goals and results and ice-break questions on sustainability (25 min); (ii) presentation and debate of the EUSTEPs university footprint calculator (conceptual structure, e-platform, and data required) (20 + 15 min); (iii) individual work on the university footprint calculator (USTEPs university footprint calculator. Available at: https://www.eusteps.eu/resources/university-footprint-calculator/ (accessed on 3 October 2023)) followed by a group exercise (60 min). All the participants had the opportunity to work individually with the calculator,

resorting to a database previously prepared by the project team using fictitious university data. This activity provided a moment of discussion among the participants, analyzing the main dimensions and categories under analysis using the university footprint calculator and their impact. Then, the participants were divided into groups, and each group was assigned to each specific parameter of the calculator, with the aim of discussing and experimenting with the impact of some activities in this parameter inside the university (e.g., in the food category, participants were able to observe how reducing fish consumption could impact the ecological footprint of their institution). The remainder of the workshop was structured as follows: (iv) presentation of the UAV and UAb ecological footprint results (10 min); (v) general debate with all the participants, relating to the challenges of data collection and the use of the e-platform; recommendations to reduce both universities' ecological footprints (60 min); and (vi) conclusions and next steps (10 min).

From the perspective of the training course, three main points for efficient data management of the calculator were highlighted, given the role of non-academic staff in this process. Gathering data from multiple sources and databases within each HEI is a challenging task, also because it requires coordinating the data collection process with different offices, departments, and individuals. The main debate was centered on three points: who should be responsible for the collection of the calculator data, what data should be collected, and how to systematically analyze the EF results in the long-term for strategic sustainability institutional transition purposes.

*4.2. Data Collection and Analysis Process*

In the first training course, two sets of questionnaires were developed and given to the participants (Table 2): (i) a pre-questionnaire to evaluate their pre-knowledge about sustainable development (concept and definition), SDGs, and EF concept, as well as to determine their expectations (applied after the first synchronous session and before starting the e-learning course lessons and e-activities), and (ii) a post-questionnaire to evaluate the participants' perceptions on the training and their overall feedback and satisfaction after the course.

The pre-questionnaire included seven close-ended questions, mainly extracted from two previously validated questionnaires: (i) four questions about the concept of sustainability and SDGs were extracted from SULITEST (the sustainability literacy test) [30]; (ii) two questions about the EF concept were extracted from the previously applied and validated three-tier diagnostic test [31], which was adopted and applied for assessing knowledge about the EF concept at universities; and (iii) one further question related to the participants' achievement expectations for the training. The pre-questionnaire was reviewed and validated by experts [32].

The post-questionnaire was conducted to evaluate the training characteristics perceived by the participants and the content and materials used during the training course. The post-questionnaire covered both closed and open-ended questions in six main areas: (i) the participants' socio–demographic characteristics, (ii) the training characteristics, (iii) the training topics, (iv) the educational materials, (v) the participants' future attitudes, and (vi) their level of satisfaction with the training course and their overall feedback. The project experts' review of the questionnaire ensured the validity of the post-questionnaire. A five-point Likert scale was applied [33] from 1 (very low) to 5 (very high) for the close-ended questions of the post-questionnaire. Descriptive data analysis was then carried out on the post-questionnaire to analyze the data [34] based on the average percentage of the responses in each question, and by using the Statistical Package for Social Sciences (v. 26 IBM SPSS Statistics, 2020).

In the second training course, only a post-questionnaire was applied, following the same goals and structure of the first training course post-questionnaire.

## 5. Results and Comparative Analysis

### 5.1. Results of the First Training Course

5.1.1. Pre-Evaluation and Characterization of the Participants' Sample for the First Training Course

Twenty-seven non-academic staff members participated in the training course from both universities. Of these, 56% were from the University of Aveiro (UAV) and 44% were from Universidade Aberta (UAb) (Table 4). All the training participants filled out an online version of the pre-evaluation questionnaire before starting the training. Most participants were female (67%), and nearly half (48%) were between 41 and 50 years. Also, as shown in Table 4, the participants had different roles and positions in different services of both universities (most were technical staff (59%), followed by administrative staff (22%)).

**Table 4.** General characterization of the participants in the first training course.

| Category | Feature | N. | % | Category | Feature | N. | % |
|---|---|---|---|---|---|---|---|
| University | UAV | 14 | 56% | Position at the university | Technical | 16 | 59% |
| | UAb | 13 | 44% | | Administrative | 6 | 22% |
| Gender | Male | 9 | 33% | | Management | 4 | 15% |
| | Female | 18 | 67% | | Others | 1 | 4% |

Regarding the participants' expected outcomes, most of them expected to increase their knowledge about 'how to calculate the EF' (87%) and 'how to be more actively involved in sustainability actions at their universities' (85%) after the course. Also, 83% of the participants expected to learn more about 'the concept of sustainability'.

Figure 1 shows the participants' pre-knowledge in three different areas of the training, including the concept of sustainability, SDGs, and EF. Based on the results of the pre-questionnaire, most of the participants had a pre-knowledge of the 'sustainability concept' and 'EF' (72% and 67%, respectively); however, the pre-knowledge approaches of 'SDGs' was less (48%). When analyzing the results of each question on the SDG topic, although the majority of the participants (74%) had a pre-knowledge of the number of goals and indicators (the first question of SDGs), when it came to the general concept of the SDGs and their focus areas (the second question of SDGs), the pre-knowledge was considerably less (only 22%).

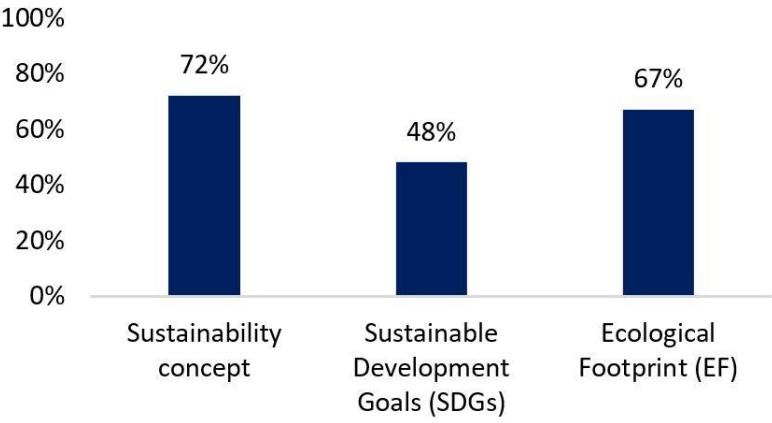

**Figure 1.** Participants' pre-knowledge on different topics of sustainability and ecological footprints.

5.1.2. Post-Evaluation Results and Overall Feedback for the First Training Course

In total, of the 27 participants who finished the training course, 20 participants returned the post-questionnaire, representing a response rate of 74%, of which 55% were from Universidade Aberta and 45% were from the University of Aveiro.

Figure 2a–e shows the participants' perceptions of the different training contents and materials. Most of the participants conveyed that the characteristics of the training were appropriate (>84%) (Figure 2a). Even though there were no remarkable differences in the participants' perceptions of the training characteristics, their first choice was the "interesting and applicable topics". Most of the participants also appreciated both the asynchronous and synchronous teaching methods, which means that the implemented teaching methods were appropriate for the training.

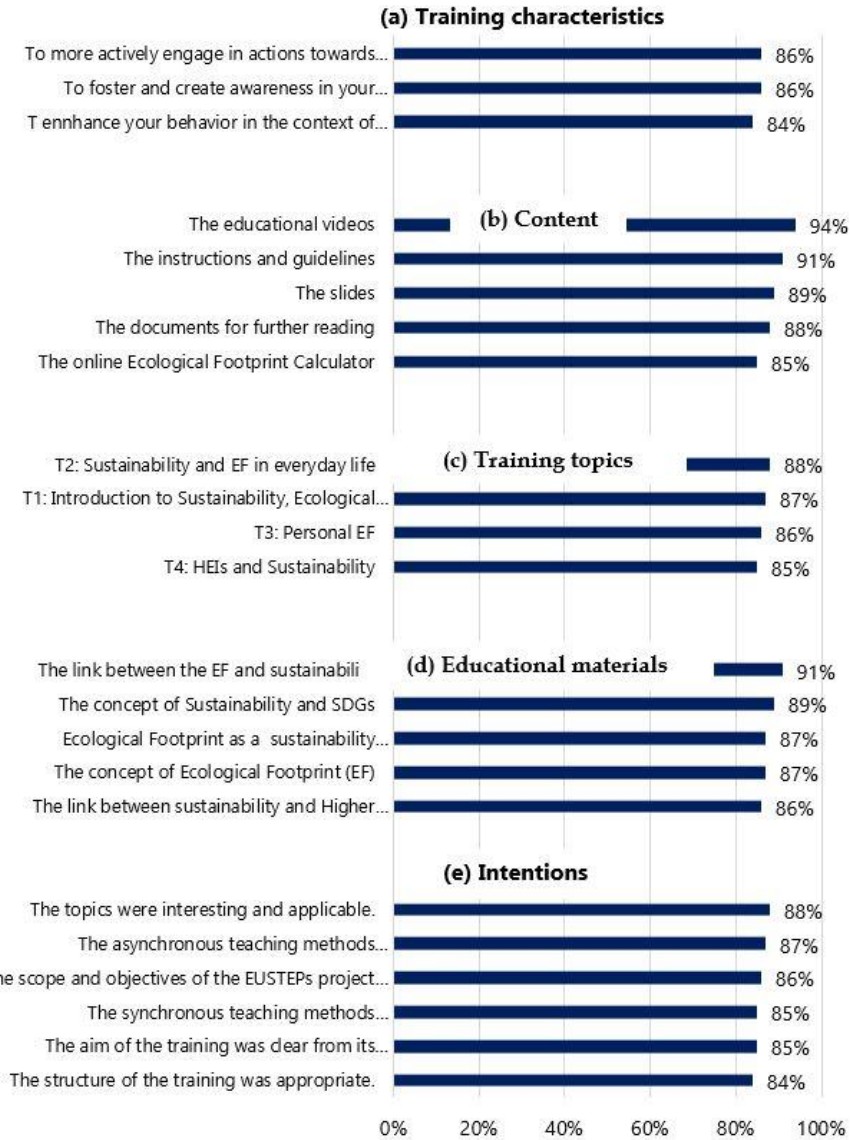

**Figure 2.** Participants' perception of the training characteristics (**a**); understanding of the topics covered by the training based on the content (**b**); understanding of the topics covered by each session (**c**); the usefulness of the educational materials used in the training (**d**); and the participants' intentions to take future action on sustainability paths (% of scoring scale) (**e**).

Also, the participants' perceptions of all training topics (>85%) indicated that the training was successful in increasing their understanding of all the topics addressed throughout the course (Figure 2b,c). However, increased knowledge was mostly observed on 'the link between EF and sustainability' (Figure 2b), where most participants stated that the training helped them to understand how EF and sustainability is linked to everyday life (Figure 2c).

All the educational materials were perceived by the participants as very useful (Figure 2d), particularly the 'educational videos', closely followed by the 'instructions and guidelines'.

Also, the results showed a high rate of increase in the participants' intentions to take future action on sustainability paths in both social and personal aspects (Figure 2e). Overall, almost all the participants were satisfied with the training (45% of participants were 'very satisfied' and 45% were 'satisfied').

Based on the participants' statements on the open-ended question, they particularly appreciated the dynamics of the educational materials, conveying that using different types of materials and activities including videos, slides, and the personal EF calculator increased their awareness regarding the impact of their professional activities on the planet. Some participants also appreciated the "dynamic breaks" which were included in the e-learning lessons as healthy physical activities to break the intensive course.

Relating to the drawbacks of the training course, a "lack of time" and "lack of synchronous sessions" were highlighted as the main drawbacks that need to be considered in the refinement of the training course. A few participants also mentioned the difficulty in forming the groups for the collective activities due to the wide variety of services/departments in which the participants worked, and some of them proposed increasing the time devoted to the training course to allow deepening of the discussion. "Replicability of the training to other participants" and the "involvement of senior university managers in the last synchronous session" were some of the suggestions made to foster the implementation of the course outcomes in practice.

*5.2. Results of the Second Training Course*

5.2.1. Characterization of the Participant Sample for the Second Training Course

The second training course included 17 participants from both universities, although only 10 participants answered the post-evaluation questionnaire (Table 5).

**Table 5.** General characterization of the participants in the 2nd training course.

| Category | Feature | N. | % | Category | Feature | N. | % |
|----------|---------|-----|-----|----------|---------|-----|-----|
| University | UAV | 10 | 59% | Position at the university | Technical | 12 | 71% |
| | UAb | 7 | 41% | | Administrative | 2 | 12% |
| Gender | Male | 5 | 29% | | Management | 2 | 12% |
| | Female | 12 | 71% | | Others | 1 | 5% |

As in the first training, most of the participants were female (71%). The distribution of age was heterogeneous. Unlike the first training, 70% were senior technicians.

5.2.2. Post-Evaluation Results and Overall Feedback for the Second Training Course

Figure 3a,b shows the participants' perceptions of the second training's efficiency and effectiveness. As previously mentioned, the second training was shorter, more practical, and allowed the participants to work with EUSTEPs' university footprint calculator and its connection with the participants' universities.

Most of the participants who completed the post-questionnaire understood the link between the EF tool and assessing the sustainability of its institution, the link between sustainability and their institution, and the link between EF and sustainability (Figure 3a). Only one participant neither agreed nor disagreed.

Most of the participants (90%) agreed or totally agreed with the need to be more actively involved in collective sustainability actions at their universities. Also, most of the participants (80%) agreed or totally agreed that they increased their intention to improve their individual behaviour within the scope of sustainability.

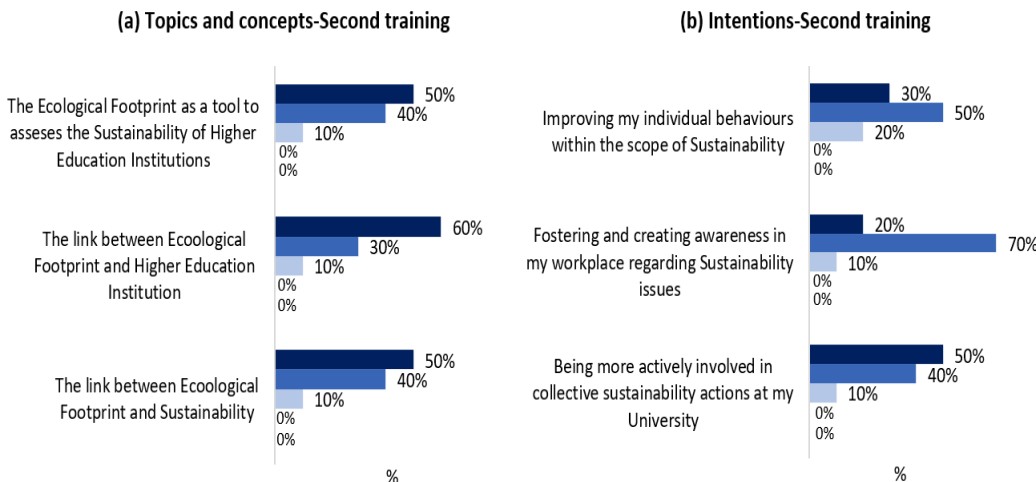

**Figure 3.** Participants' (**a**) understanding of the topics and concepts covered by the training based on the content; (**b**) understanding of the importance of the topic covered by each session (intentions).

Regarding the evaluation of the different features of EUSTEPs' university footprint calculator (Figure 4), the participants' feedback was positive across all the topics. Most of the participants (60%) considered the calculator's design to be very good. Regarding the perception of the user-friendliness of the calculator, most of the participants expressed that it was very good or good (60% and 20%, respectively) and 40% considered the user-friendliness as average. Relating to the footprint results interface on the calculator ('analysis of results'), most of the participants expressed it was very good or good (50% and 30%, respectively). When asked about their satisfaction with the second training course, 60% were very satisfied and 40% were extremely satisfied.

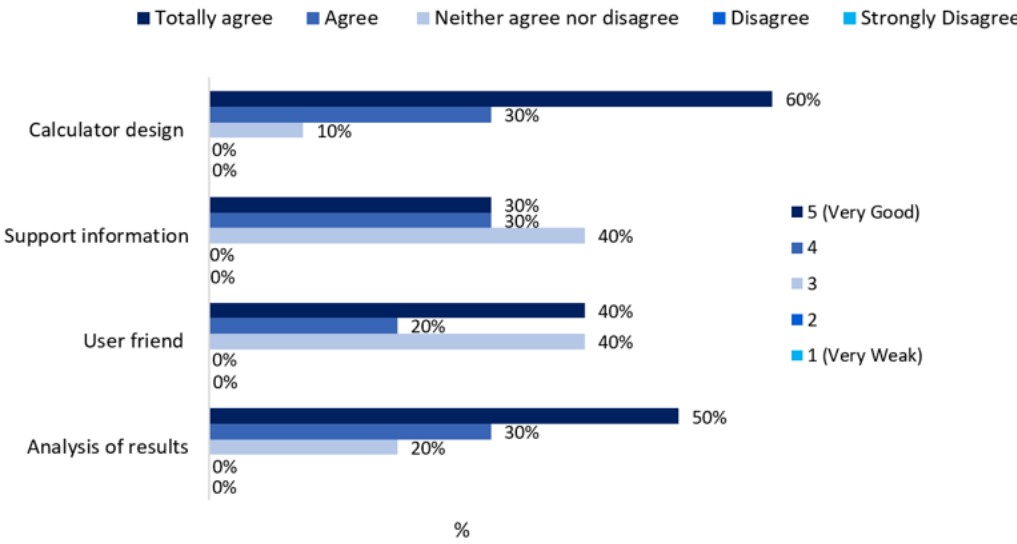

**Figure 4.** Participants' perception of the evaluation features of EUSTEPs' university footprint calculator.

## 6. Discussion

When comparing the pre- and post-learning outcomes (Figure 5) of the first training course, all of the assessed parameters were perceived with greater achieved knowledge than the initially expected outcome (>86%). Even though there was a small difference between the five assessed parameters of achieved knowledge (based on the participants' perceptions), the perception of increased knowledge was greater relating to 'the concept of sustainability and the SDGs' (89%). This finding was also confirmed by one of the participants' statements:

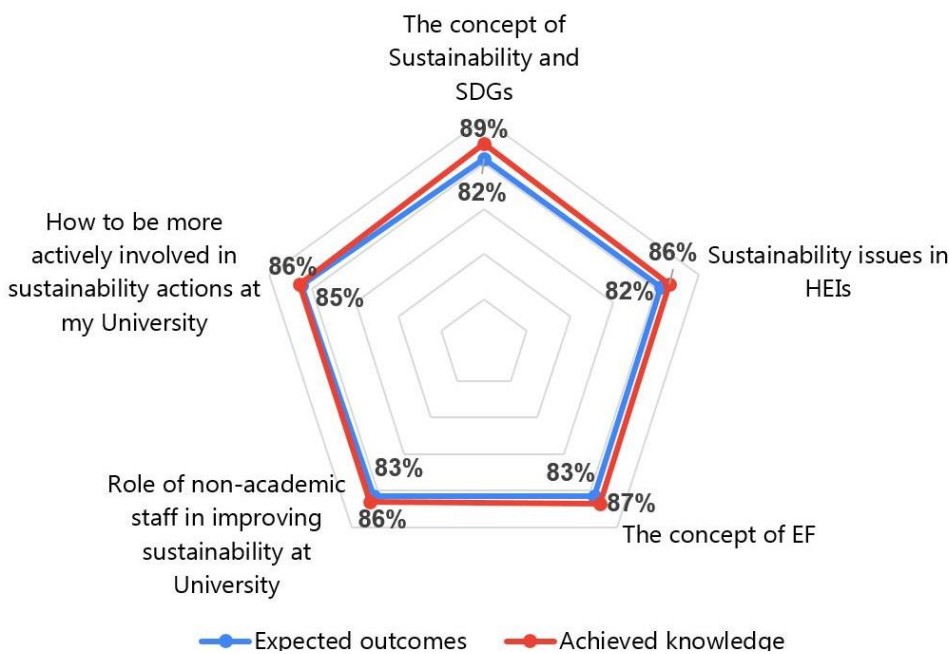

**Figure 5.** Achieved knowledge versus expected outcomes relating to different topics in the training course.

"I will use the bicycle and public transport much more, instead of the comfort of the car, and I will try to buy foodstuffs from street producers".

Hene, our results suggest that the first training course was successful in increasing sustainability knowledge in the workplace, and also in increasing the participants' intentions to improve their sustainable behaviours in daily life. Also, based on the results of the pre-test, in which the lowest score for pre-knowledge was on the 'concept of the SDGs' (48%, Figure 1), the training was fully successful in helping non-academic staff to increase their knowledge about the SDGs (Figure 5). However, as reported by various authors e.g., [4], the process of creating awareness and generating effective change is not immediate, and it takes time for the outcomes to be implemented.

Increasing individual knowledge about sustainability and SDG issues as education and research elements at HEIs could also help to improve the sustainability features of other core elements of HEIs such as governance, operations, assessment, and reporting. Similarly, Leal Filho et al. [5] also reported that embedding the SDGs within and across courses would contribute to extending human capital, yield an increase in the number of people acting and aiming to live sustainably, and have a significant impact on securing the achievement of the goals and a better future. This issue was clear in one of the participants' comments in the open-ended question:

"It is extremely important that everyone is aware that the little they can contribute counts".

HEIs are ethically and morally responsible for increasing the awareness, knowledge, skills, and values that are needed to create a more sustainable way of living [35]. In doing so, they should also and fundamentally involve all members of the university community, including non-academic staff, in the co-development process of building a sustainable university. To do so, 'the presence of senior managers or a person in charge of the HEI's management body during the training course' was referred by several participants as a future development direction for the training course in order to share and involve both academic and non-academic communities in this topic.

The participants' opinions on how the calculator can be used and/or enhanced at their universities were collected using the open-ended questions on the post-questionnaire. Based on their feedback, the university footprint calculator is a fundamental tool that can (i) be applied in different areas of activity of universities, including a strategic area to define objectives, (ii) prioritize the sustainable measures at the university that are financially

possible to implement, (iii) trigger the involvement of the academic community by fostering a sense of active civic intervention and pride of belonging to the university, (iv) integrate the results and demonstrate work that is already being developed by universities, and (v) demonstrate the importance of involving the rectory in the process of calculating and reducing the EF of the university.

However, the variety in the participants' educational backgrounds, different academic departments, and careers was pointed out as a challenge by one participant. Also, two participants from Universidade de Aveiro referred to a greater challenge associated with learning through an e-learning environment in comparison with face-to-face training. This is a commonly reported challenge for first-time students participating in e-learning courses, which is usually overcome successfully by a 5 to 7 days asynchronous boot camp in larger courses [27]. In our short 11 days of training (first round), the course design involved two synchronous moments, with the first one also serving as a boot camp. Future training courses will need to program the first synchronous one-hour session in order to avoid the reported difficulties. Also, although the results revealed knowledge increases after the course, these were based on the participants' perceptions. Therefore, future training courses, should additionally consider the inclusion of more precise assessment tools for knowledge acquisition (such as quizzes), in addition to the participants' perceptions, which could enrich the study and future developments of the training course, as also emphasized by Perbandt et al. [27]. Furthermore, the participants should be post-assessed sometime after the training to allow for evaluating their behavior changes and to assess their contribution to the suitability integration at their institution.

According to United Nations recommendations, there is a need to create an intermediate structure to foster change towards accelerating sustainability implementation by using a two-way approach which involves the top leadership and the students, university staff, and society, and which can be achieved in a collaborative manner [36]. According to this two-way approach of the United Nations, within the sustainable offices of both universities and based on this experience (and recommendations by the participants in the post-questionnaire), future regular training programs for non-academic staff could be planned with key actors during the last stages of the training. Their aim will be brainstorming about workplace improvements and raising the awareness of the whole university community on the availability and applicability of the calculator, in collaboration with the top managers, teaching staff, and students, benefiting from their different roles at the universities as well as exchanging the experiences of both universities. Earlier similar experience involving a one-round workshop with Universidade Aberta has shown successful results [37].

## 7. Conclusions

Higher education institutions are home to very large communities. Non-academic staff play a crucial role in the everyday functioning of these institutions, and therefore are key agents in universities' actions. Training this staff is vital to ensure lifelong learning processes and a well-informed body of staff. Sustainability training is included in this special asset, set since HEIs are becoming more aware of their impact on the surrounding environment and are working towards more sustainable campuses. In order to ensure this goal, every person in a university's community should be informed on sustainability issues, including academic staff and the students, as well as non-academic staff. This training field is yet to be explored, since there are few reports on implementing staff training, particularly in sustainability.

The project successfully developed two online training courses to target the non-academic staff of the two Portuguese universities. These courses had a positive impact in two ways. Firstly, they increased the self-perceived knowledge of non-academic staff on sustainability and EF issues. Secondly, the course empowered them with participatory skills to transform the workspace and the whole university community towards sustainability. The training was a complete success in increasing the commitment of non-academic staff

to engage in future actions towards sustainability. This commitment was not limited to their work positions at universities, and could be also extended to their everyday lives. Overall, the training courses played a key role in motivating the staff to actively contribute to sustainable practices, both on an individual and collective level.

By presenting and discussing sustainability within the context of everyday life rather than through abstract concepts on sustainability, and by bridging it with the 2030 UN Agenda Sustainable Development Goals, the non-academic staff of both universities better understood how sustainability relates to the entire spectrum of daily life activities, including their workplace and the administration of HEIs, as well as how to contribute to the implementation of sustainability practices at their HEI.

The results also showed that the use of different dynamic educational materials, particularly videos, slideshows, and the EF calculator activity, helped to increase knowledge on the sustainability issues under study among the non-academic staff. Among the course topics, Topic 2 (EF and sustainability in daily life) and the associated link between sustainability and EF were perceived as fundamental, helping the participants to increase their awareness of their personal EF and their impact on professional activities and on the planet. As a result of these training courses, some of the non-academic staff willingly engaged in the data collection phase for the calculation of the university EF to assess the current impacts of the university activities and services on their EF and help to reduce its impacts.

The main limitation of this study was its relatively small sample size (in the first training course, n = 27, and in the second training course, n = 17), as well as the unbalanced number of participants in both training courses, which may lead to lower statistical significance. However, as stressed by Perbandt et al. [27], small-sample studies can still obtain meaningful results if well designed, provided careful interpretation.

Other higher education institutions can use the trainings courses (online or face to face) and the approach that were developed in this research to empower non-academic staff for the implementation of sustainability, hence contributing to the whole school and the successful integration of sustainability within HEIs.

**Author Contributions:** Conceptualization, P.B.-N. and S.C.; Methodology, P.B.-N. and S.C.; Formal analysis, M.M. and M.N.; Investigation, M.M.; Writing—original draft, M.M. and M.N.; Writing—review & editing, P.B.-N., S.C., S.M.P., C.M. and M.F.D.; Visualization, A.P.G., M.L. and H.N.; Supervision, P.B.-N.; Project administration, S.C.; Funding acquisition, G.M. All authors have read and agreed to the published version of the manuscript.

**Funding:** This research was funded by Erasmus+ program of the European Union, grant number [2019-1-EL01-KA203-062941]. The work was also supported by Fundação para a Ciência e Tecnologia, IP, Portugal: CENSE (UID/AMB/04085/2019); GOVCOPP (UIDB/04058/2020) + (UIDP/04058/2020); CESAM & DEP (UIDP/50017/2020 + UIDB/50017/2020), and CFE (UID/BIA/04004/ 2020).

**Institutional Review Board Statement:** Not applicable.

**Informed Consent Statement:** Informed consent was obtained from all subjects involved in the study.

**Data Availability Statement:** Data sharing is not applicable to this article.

**Acknowledgments:** The EUSTEPs team acknowledges all the participants who attended both training sessions. The team also acknowledges the contributions of Ana Isabel Miranda and Filipe Teles to the project.

**Conflicts of Interest:** The authors declare no conflict of interest.

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
