# Peer review of "Empowering Non-Academic Staff for the Implementation of Sustainability in Higher Education Institutions"

_sustainability, doi:10.3390/su152014818_

Round 1
Reviewer 1 Report
Training for university staff seems valuable for individual participants personally and for the university as a community itself and as an exemplar of how towns and cities can learn and change. I think this training is a fine idea, and I'm happy to see how these universities are approaching the task. The materials and practice that the authors have established at their respective institutions seem highly useful for an ongoing, university-wide effort to inform an HEI community about sustainability and to establish practices throughout the HEI that are sustainable. Consequently, it seems to me that the most notable outcome of the training is the establishment of a precedent for it to occur on an annual basis, attracting more and more staff members to it over time so that the whole university adopts sustainable goals and methods. I wonder if the authors might request that university leaders incentivize the training for all staff members somehow. If staff learn about sustainable practices, they can help create (or maintain) more sustainable ones than are in current use in their own individual work roles and spaces. For example, administrative assistants can encourage waste reduction (particularly paper waste), and groundskeepers can work to reduce water usage, runoff, and erosion. For these changes to occur, more staff members (and their supervisors) need to understand sustainability and sustainable work practices. I wonder if the authors might be more explicit in their discussion of plans for ongoing training and institution-wide change. Also, maybe training programs for technical staff members could occur within specialty groups so that brainstorming about workplace improvements could occur during the last stages of the training program? I wonder if the authors may want to mention some of these possibilities.
Despite the value of the training, the description of the results is sometimes less precise than it should be. Given that the results are from two opinion surveys, I think the authors should take more care than they currently do in claiming that the data show success. They suggest it, but I don't think they "mean" it, as was stated on lines 472 and 553.
Some of the figures in the article need revision to eliminate reading difficulty or confusion. Table 1 is OK except for the row about "Primary functions." In the right-hand column, I don't know what "powers" the authors are talking about. Can the meaning here be clarified? Table 3 needs refinement with the width of the columns and the spacing after bullets. Figure 2 is inelegant and visually misleading. The bars in "a" are all much longer than in the other categories. I think it would be better to display all categories (a-e) in one combined horizontal bar graph with divisions between categories. The shaded legends in Figures 3 and 4 are difficult to distinguish. I encourage the authors to use direct labeling and eliminate the legends altogether. Figure 6 is not effective as a figure to me. I wonder if the authors are quoting participants' comments or interpreting them from the survey in this figure. The source of this information should be more apparent. Also, I think that the information itself is not suitable for a figure. It would be better presented in text paragraphs with bullet points. In a textual presentation, the authors could clarify the source of the comments, cause and effect assertions, and ideas for further development.
The authors need to review their use of "this" pronouns throughout the article to ensure that when they use "this" as the subject of a sentence it has a clear, identifiable noun antecedent. Instances of ambiguous pronouns appear on lines 472, 487, 517, and 521. Additional problematic pronouns may also appear earlier in the article; the whole should be checked.
Capitalization also needs scrutiny throughout. Some words are capitalized for no apparent reason. See line 533: should the line read "life-long knowledge," or are the authors referring to a proper noun?
Though the authors' meaning is usually clear in expression, some confusing statements exist, and often they appear at critical junctures. Here is a list of sentences that merit more precise phrasing:
lines 455-56: I'm not sure what this sentence means. Has the calculator analyzed the EF results for each participant? Do the ratings indicate whether participants understand how the results of the calculator occurred? Phrase this point more precisely.
line 479-80: Odd, circuitous phrasing. Can you state this point more directly? When the final clause begins, the sentence loses its clarity. A comma after "and" would help, but the phrase "impact of the outcomes" is abstract, I would replace it.
lines 494-96: The syntax and punctuation of this sentence are unclear. Who is doing the recognizing? Should the verb for "knowledge, skills, and values" be "are needed"? Is that the point, or are additional words missing?
Author Response
Dear Reviewer 1,
We are gratefull for all your comments.
All your suggestions were taken into account to improve the article.
We answered to all the points mentioned in the atttached file 'Response to Reviewer 1' (copy below).
Kind regards,
Paula Nicolau
-----
Response to Reviewer 1 Comments
Point 1: I wonder if the authors might be more explicit in their discussion of plans for ongoing training and institution-wide change. Also, maybe training programs for technical staff members could occur within specialty groups so that brainstorming about workplace improvements could occur during the last stages of the training program? I wonder if the authors may want to mention some of these possibilities.
Response 1: We thank Reviewer 1 for the valuable suggestion. In order to be more explicit about the institution's future plans for the training programs targetting technical staff members, we have included a paragraph (lines 537-546) in Discussion section.
Point 2: Despite the value of the training, the description of the results is sometimes less precise than it should be.
Response 2: The manuscript, and particularly the Results section, was throughly checked to make the text clearer.
Point 3: Given that the results are from two opinion surveys, I think the authors should take more care than they currently do in claiming that the data show success. They suggest it, but I don't think they "mean" it, as was stated on lines 472 and 553.
Response 3: Our feeling of "success" is due to: (i) the expression of surprise and sometimes shock with the novelty of the topics on sustainability and SDGs, and the use of the Footprint calculator (during the e-ativities of the training), (ii) the enthusiam and commitment of the participants during the last synchronous session, (iii) the enthusiam and commitment during the course assessment, and (iv) later on, after the course finished, with their commited participation during the the first calculation of the University Footprint. We have changed the text on lines 556-557 to emphasize the later point.
Point 4: Some of the figures in the article need revision to eliminate reading difficulty or confusion. (i)Table 1 is OK except for the row about "Primary functions." In the right-hand column, I don't know what "powers" the authors are talking about. Can the meaning be clarified? (ii) Table 3 needs refinement with the width of the columns and the spacing after bullets. (iii) Figure 2 is inelegant and visually misleading. The bars in "a" are all much longer than in the other categories. I think it would be better to display all categories (a-e) in one combined horizontal bar graph with divisions between categories. (iv) The shaded legends in Figures 3 and 4 are difficult to distinguish. I encourage the authors to use direct labeling and eliminate the legends altogether. (v) Figure 6 is not effective as a figure to me. I wonder if the authors are quoting participants' comments or interpreting them from the survey in this figure. The source of this information should be more apparent. Also, I think that the information itself is not suitable for a figure. It would be better presented in text paragraphs with bullet points. In a textual presentation, the authors could clarify the source of the comments, cause and effect assertions, and ideas for further development.
Response 4: We agree with Reviewer 1. Tables and figures were changed accordingly: (i) In Table 1, the text was changed to "Functions and competences (...)"; (ii) Table 3 was improved, concerning the "refinement with the width of the columns and the spacing after bullets"; (iii) Figure 2 was changed to a combined horizontal bar graph (lines 402-403) (iv) The bar colour shades in Figures 3 and 4 were changed to improve their “readability”, and onde common language was added, (v) Figure 6 was removed and instead, a paragraph was added that covered the source of information for the participants' opinions, as well as the opinions on how the Calculator can be used/enhanced (lines 506-515). The suggestions for future development have been integrated into the proposals for the future plan in the last paragraph of the Discussion (lines 537-546; highlighted green).
Point 5: Relating Comments on the Quality of English Language.
The authors need to review their use of "this" pronouns throughout the article to ensure that when they use "this" as the subject of a sentence it has a clear, identifiable noun antecedent. Instances of ambiguous pronouns appear on lines 472, 487, 517, and 521. Additional problematic pronouns may also appear earlier in the article; the whole should be checked.
Response 5: All "this" pronouns were checked through the text (including those in lines 472, 487, 517, and 521), and changed were needed to ensure clarity of the text (all highlighted in green).
Point 6: Capitalization also needs scrutiny throughout. Some words are capitalized for no apparent reason. See line 533: should the line read "life-long knowledge," or are the authors referring to a proper noun?
Response 6: Capitalization was checked througout the article. The text “life-long knowledge” was changed to a more appropriate expression: 'life long learning’.
Point 7: Though the authors' meaning is usually clear in expression, some confusing statements exist, and often they appear at critical junctures. Here is a list of sentences that merit more precise phrasing (…).
Response 7: The authors agree with the reviewers commnent relating to the “confusing statements”.
Hence, the text contained in lines 455-456; lines 479-480, and lines 494-496, was re-writen to make it clearer. In somecases, the text prior or after the above mentioned lines, was also re-writen for clarity. All the changes were highlighted in green.

Reviewer 2 Report
Dear authors,
The subject of the project and the work are topical. Indeed, in terms of training non-academic staff for sustainable issues in HEIs, they are not well developed in the literature.
However, I found some aspects which need to be improved.
1. Before to use acronyms is necessary to define them. So, please revise all acronyms used in the article and replace where is necessary. For example: EF is used for the first time at line 91 and defined in line 171 and not used in lines 180, 193 and so on. Same for HEI.
2. Revise all the tables to be more attractive and well presented. Please see especially Table 3.
3. In Methods, the methods used in the article must be presented. So, please reformulate. Project stages/steps can be used in the development of method tools.
4.The reason for the lower number for the second target group is not developed. Missing 10 out of 27 people is an important issue for sustainable results.
5. The limitations of the study are not presented in the Conclusions.
6. The bibliography is not uniformly presented
Author Response
Dear Reviewer 2,
We are grateful for all your comments.
All your suggestions were considered to improve the article.
We answered to all the points mentioned in the attached file 'Response to Reviewer 2' (copy below).
Kind regards,
Paula Nicolau
--------------------------
Response to Reviewer 2 Comments
Point 1: Before to use acronyms is necessary to define them. So, please revise all acronyms used in the article and replace where is necessary. For example: EF is used for the first time at line 91 and defined in line 171 and not used in lines 180, 193 and so on. Same for HEI.
Response 1: All the manuscript was revised for the acronyms. The meaning of the acronym EF was introduced in line 91, and was highlighted in blue. The acronym of HEI was already introduced in line 47.
Point 2: Revise all the tables to be more attractive and well presented. Please see especially Table 3.
Response 2: All tables were revised to be more attractive and better presented, particularly Table 3, where column width adjustement and bullets format were improved.
Point 3: In Methods, the methods used in the article must be presented. So, please reformulate. Project stages/steps can be used in the development of method tools.
Response 3: We have reestructured the text to make clear the information relating to the Methods. Hence, (i) we have shifted the sub-section on "The EUSTEPs project and the universities involved" from the methods to a new section; currently, it is section 3 (lines 155-205);
(ii) The Methods section is currently section 4 (lines 206–355) and is structured as follows:
4.1. "The short-term sustainability training courses to non-academic staff";
4.2. "Data collection and the analysis process".
Point 4: The reason for the lower number for the second target group is not developed. Missing 10 out of 27 people is an important issue for sustainable results.
Response 4: We appreciate your valuable comment, and have included a justification text on page 6 (lines 215-222), after Table 2 (highlighted in blue colour).
Point 5: The limitations of the study are not presented in the Conclusions.
Response 5: We thank reviewer 2 for valuable comment, and we have included the study limitations on the Conclusions section (page 16, lines 578-587; higlighted in blue colour).
Point 6: The bibliography is not uniformly presented
Response 6: Bibliography was fully revised for uniformity.

Reviewer 3 Report
Many thanks for the submission to the Sustainability journal. I believe that it is an important and timely manuscript/topic.
However, please read the comments below carefully, towards strengthening the manuscript further:
- I believe that the title could, or should, indicate that this study is within the context of Portugal. This would alert the reader to it regarding universities from that country.
- The contributions, both theoretical and practical/policy, should be outlined within the introduction more clearly. This will highlight the value of the manuscript with greater explanation.
- I believe that a brief section, from the beginning of the literature review, should address the concept of sustainability and discuss how the university environment engages with existing measures and projects etc. Before non-academics are discussed specifically.
- Furthermore, a section discussing the context of Portugal, discussing the publications or government intervention from academic and industry perspectives concerning sustainability would be beneficial here.
- I believe that greater justification of the method applied here should be given. Why this method selected, and not other methodological approaches? Why is this more suitable?
- I would suggest that 2 to 3 overarching, and fundamental themes, realised from the data collection, should form the sub-sections of the discussion section.
- The conclusion section, I believe, should be clear in providing implications and recommendations for policy sub-sections to revisit the aim of the manuscript and provide precise comments and closing points for the reader and relevant stakeholders.
Author Response
Response to Reviewer 3 Comments
The authors are grateful for Reviewer’s 3 comments and suggestions.
Point 1: I believe that the title could, or should, indicate that this study is within the context of Portugal. This would alert the reader to it regarding universities from that country.
Response 1: We thank reviewer 3 for his/her suggestion, which we had also considered while preparing the manuscript. However, this paper reports on a study case with materials and methods developed with other European universities (within an Erasmus+ project) ... which happened to be tested in two Portuguese universities. It is not a general study for Portugal, as is indicated in the Summary section. The Introduction, presents the worldwide issue of Empowering non-academic staff for the implementation of sustainability in higher education institutions, and subsequently the study presents the results of an original study, that was tested in two HEI in Portugal. Hence, we do not feel comfortable to include the reference of Portugal in the title. However we value the suggestion and the word “Portugal” was added to the keywords.
Point 2: I believe that a brief section, from the beginning of the literature review, should address the concept of sustainability and discuss how the university environment engages with existing measures and projects etc. Before non-academics are discussed specifically.
Response 2: We are grateful to the valuable suggestion. A section was inserted at the beginning of the section 2. (on ‘Review sustainability implementation with non-academic staff’) addressing the concept of sustainability and discussing how the university environment engages with existing measures and project, etc. (see text highlighted in green).
Point 3: Furthermore, a section discussing the context of Portugal, discussing the publications or government intervention from academic and industry perspectives concerning sustainability would be beneficial here.
Response 3: A section discussing the context of Portugal, including government and academic intervention concerning sustainability was added in section 2. (on ‘Review sustainability implementation with non-academic staff’) (see text highlighted in green).
Point 4: I believe that greater justification of the method applied here should be given. Why this method selected, and not other methodological approaches? Why is this more suitable?
Response 4: We agree with Reviewer. The applied method was better justified in the Methods section (see text highlighted in green).
Point 5: I would suggest that 2 to 3 overarching, and fundamental themes, realised from the data collection, should form the sub-sections of the discussion section.
Response 5: Again, we thank Reviewer 3 for the valuable suggestion. The discussion section was designed, aiming to integrate all the main themes of the research and its results. If we divide the Discussion further into sub-sections, the text would be much longer, and loose its focus on the results integration.
Point 6: The conclusion section, I believe, should be clear in providing implications and recommendations for policy sub-sections to revisit the aim of the manuscript and provide precise comments and closing points for the reader and relevant stakeholders.
Response 6: The conclusion section was reestructured, in order to make clear what were the main conclusions of study, main limitation and pointing to future studies. Also, some extra sentences were added (see text highlighted in green).
Point 7: Table comment | Are all the cited references relevant to the research? Can be improved. | Is the article adequately referenced? Can be improved.
Response 7: All references were checked for relevance. Three references were added related to points improved above (highlighted in green).

Reviewer 4 Report
The manuscript had a importance for scientific community and it refers to the implementation of sustainability in higher education institutions.
The title of the article is suitable and I consider subsections and structure of the manuscript appropriate.
The quality of English language is adequate. The references are sufficient.
Figure 3 and Figure 4 exceed the edges on the left. They will have to be rearranged on the page.
Overall, it is a good article, with minor changes can be published in the Sustainability.
The quality of English language is adequate.
Author Response
Response to Reviewer 4 Comments
The authors are grateful for Reviewer’s 4 comments.
The text was thoroughly revised for English.
Point 1: The clarity of the research design, questions, hypotheses and methods can be improved.
Response 1: The clarity of the research design, questions, hypotheses and methods were improved in the methods section (text highlighted in green).
Point 2: Clarity of the presentation of results from empirical research can be improved.
Response 2: All the results section was reviewed and changes were made to improve clarity of the presented results.
Point 3 (comments and suggestions): The manuscript had a importance for scientific community and it refers to the implementation of sustainability in higher education institutions.
The title of the article is suitable and I consider subsections and structure of the manuscript appropriate.
The quality of English language is adequate. The references are sufficient.
Figure 3 and Figure 4 exceed the edges on the left. They will have to be rearranged on the page.
Overall, it is a good article, with minor changes can be published in the Sustainability.
Response 3: The authors are grateful for the comments of Reviewers 4.
Figures 3 and 4 were rearranged on the page and kept within the page edges.

Round 2
Reviewer 3 Report
Thank you for your resubmission to the Sustainability journal.
I believe that the previous reviewer comments have been carefully considered, with new additions / edits to the manuscript being relevant and justified.
I am happy to recommend an accept decision, at this stage.